# Rapid disintegration and weakening of ice shelves in North Greenland

R. Millan [1] ✉, E. Jager[1], J. Mouginot [1], M. H. Wood [2], S. H. Larsen [3], P. Mathiot [1], N. C. Jourdain [1] & A. Bjørk [4]

The glaciers of North Greenland are hosting enough ice to raise sea level by 2.1 m, and have long considered to be stable. This part of Greenland is buttressed by the last remaining ice shelves of the ice sheet. Here, we show that since 1978, ice shelves in North Greenland have lost more than 35% of their total volume, three of them collapsing completely. For the floating ice shelves that remain we observe a widespread increase in ice shelf mass losses, that are dominated by enhanced basal melting rates. Between 2000 and 2020, there was a widespread increase in basal melt rates that closely follows a rise in the ocean temperature. These glaciers are showing a direct dynamical response to ice shelf changes with retreating grounding lines and increased ice discharge. These results suggest that, under future projections of ocean thermal forcing, basal melting rates will continue to rise or remain at high level, which may have dramatic consequences for the stability of Greenlandic glaciers.

The Greenland ice sheet has contributed 17.3% of the observed rise in sea level in the period 2006–2018, and has thus become the second largest contributor after ocean thermal expansion[1–3]. During the last forty years, mass losses from the ice sheet have increased from near-balance to a loss rate of 286 ± 20 Gt/yr in 2010–2018, with 66% being attributed to glacier dynamics and 34% to increased surface melt[4,5]. Recent studies have shown that the intrusion of warm Atlantic water was responsible for widespread enhanced calving rates at marine terminating glaciers around Greenland[6,7]. Mass losses increased more or less simultaneously in the northwest, southeast and centralwest part of the ice sheet during the 80s and the 90s[4,8]. However, glaciers in North Greenland only started to be out of balance after 2000[9], due to changes in the floating extension (ice shelves) of a couple of glaciers[4,10,11]. In 2018, the mass losses of these glaciers due to ice discharge remained however moderate compared to the other sectors of the ice sheet (65.4 ± 3.3 Gt/yr vs 124 ± 3.5 Gt/yr for northwest, 165 ± 6 Gt/yr for southeast in 2018)[2,4].

Overall, 25% of the ice sheet area is drained through former or remaining ice shelves, which represents a sea level rise equivalent of 2.1 m[4,5,8,12]. If the glaciers located in North Greenland lose the buttressing provided by ice shelves, the increase in discharge[10,13,14] could rival the largest contributors to Greenland ice mass loss (e.g., southeast and northwest). Events such as the collapse of Zachariæ Isstrøm in 2003, the large calving event at Petermann in 2012 or the thinning of the 79 N ice shelf already triggered increase in dynamic mass losses[10,11,15,16]. Despite their fundamental buttressing role, there is to date no comprehensive overview of these ice shelves evolution, which hampers our ability to understand the processes leading to their weakening and collapse, and their relation with glacier mass changes. It is thus extremely important to define the timing and drivers of historical and current changes of ice shelves, as well as glacier response, in order to better predict the contribution of Greenland to sea level rise.

In this study, we provide a long-term and holistic view of ice shelf evolution in North Greenland. The eight ice shelves that are surveyed are the floating extensions of the following glaciers: Petermann (38 cm Sea Level Equivalent - SLE), Steensby (1.4 cm), Ryder (13 cm), Ostenfeld (3.9 cm), Hagen Bræ (6.5 cm), 79 N (60 cm), Zachariæ Isstrøm (ZI, 55 cm) and Storstrømmen/Bistrup Bræ (SB, 33 cm) (Fig. 1). We document the evolution of basal melting rates, calving fluxes, ice front/grounding line (GL) positions, ice shelf volume, velocity and discharge using a combination of multiple remote sensing datasets and outputs from a regional climate model (Methods, Supplementary Data 1).

[1]Université Grenoble Alpes, CNRS, IRD, INP, 38400 Grenoble, Isère, France. [2]Moss Landing Marine Laboratories, San José State University, San Jose, CA 95192, USA. [3]Department of Glaciology and Climate, Geological Survey of Denmark and Greenland (GEUS), Copenhagen, Denmark. [4]Department of Geosciences and Natural Resources Management, University of Copenhagen, 1350 Copenhagen, Denmark. ✉e-mail: romain.millan@univ-grenoble-alpes.fr

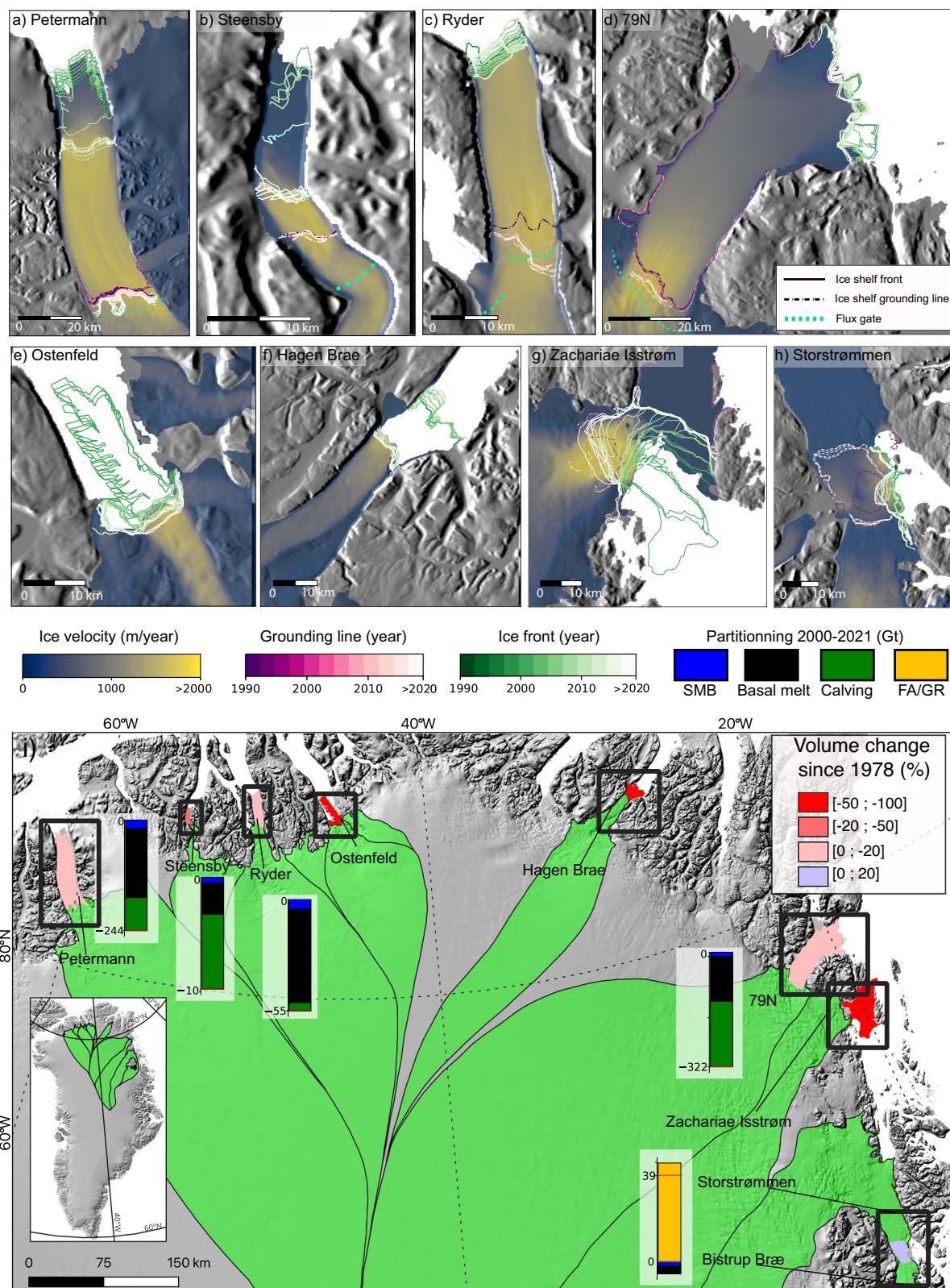

**Fig. 1 | Ice shelves changes in North Greenland.** Changes in ice shelf frontal and grounding line position between 1990 and present (**a**–**h**). Ice flow velocity (source:[4]) is color coded on a linear color scale and overlaid on a shaded version of the digital elevation model from Bedmachine v3[12] (**a**–**h**). **i** Location of all ice shelves in North Greenland with their maximum extent over the study period. Partitioning is shown as bar plot for the period 2001–2021, with Surface Mass Balance (SMB) in blue, basal melting in black and calving in green. The total cumulative mass budget for each ice shelves is noted on the bar plot in Gt (negative for mass loss). FA/GR indicates positive ice shelf mass change from calving, which is typically found when floating area increase with grounding line retreat (GR) or ice front advance (FA, see Methods). Ice shelves colors correspond to the percentage of volume change since 1978. Flux gates from[6] are shown as dotted light green lines. No grounding lines are available in 1997–2010.

Finally, we use Conductivity Temperature Depth (CTD) measurements and an Arctic Ocean Physics Reanalysis (AOPR) to compare the ice shelves' evolution with changes in ocean temperature[6,17] (Methods).

## Results

### Ice shelf changes in North Greenland

Over the eight main ice shelves, the floating extensions of Zachariæ Isstrøm, Ostenfeld and Hagen Brae completely collapsed between 2003 and 2010 (Figs. S1–S3). In 2003, 80% of the floating section of Ostenfeld collapsed, which translated into a volume loss of $27 \pm 2$ km$^3$ since 1978. Between 2001 and 2005, the ice shelf of Hagen Brae started to dislocate in the shear margins (Figs. S2–S4), and dropped from $21.8 \pm 0.6$ km$^3$ (2005) to an average of $2.7 \pm 0.6$ km$^3$ in 2009-present. The ice shelf of Zachariæ Isstrøm almost completely collapsed between 2003 and 2012[7], decreasing from $130 \pm 1$ km$^3$ to less than $25 \pm 1$ km$^3$ (Fig. 1). The GL of Ostenfeld and Hagen Brae remained stable over the entire period, while the GL of Zachariæ retreated by 7-km in 1996–2015[10] (Fig. 1).

Below we describe significant changes among the five remaining ice shelves. We observe a GL retreat for all of them, except Steensby, whose position remained stable since 1992 despite considerable terminus retreat. In August 2014, after enhanced fracturing, this ice shelf shrank to an area of 23.8 km$^2$, or 34% of its area in 2000–2013. Retreat of the Petermann's GL was reported to be 7 km between 1992 and 2021, with 5 km occurring in the last five years[18]. In 2020, the ice front readvanced close to its position prior to 2012, leading to an increase of 35% of the floating area compared to 2012–2020. At Ryder ice shelf, the western and the eastern part of the GL retreated by 2.2 and 5.7 km respectively between 1992 and 2011 (Fig. 1). Until 2020, the eastern part continued to retreat by an average distance of 2.6 km, for a total of 8.3 km since 1992, which is the largest retreat observed (Fig. 1). Ryder's GL retreat increased the ice shelf area from 245 km$^2$ in 2006 to an average of 280 km$^2$ after 2015. At 79 N ice shelf, the GL remained stable between 1992 and 2011, and started to retreat later on between 2011 and 2016 by 2.0 km (Fig. 1). This was followed by another retreat event of 1.4 km in 2019–2020, for a maximum cumulative recession of 4.2 km since 1992, which translates into a 2% increase in area compared to 2009–2010. In 2020, the northern branch of 79 N broke off completely, resulting in an abrupt 5% decrease in the ice shelf area. The GL of Storstrømmen and Bistrup Brae (which surged in 1978 and 1988) retreated by an average distance of 3.0 and 8.0 km respectively between 1996 and 2016, which is consistent with recent studies[2] (Fig. 1). Storstrømmen's GL further retreated by 2.0 km in 2016–2020, and is now within 2.8 km of its pre-surge position. The GL position for Bistrup Brae migrated upstream by 0.75 km during the same time period (Fig. 1). Consequently, the ice shelf area for SB increased from 80.6 km$^2$ in 2013 to 273.8 km$^2$ in 2018.

We show evidence of a consistent increase in basal melting below Petermann (Fig. 2a). First, basal melting rates decreased from $14.1 \pm 1.6$ m/yr in 2001–2002 to $11.8 \pm 1.1$ m/yr in 2003–2004. From 2005 to 2016, width averaged GL melt rates increased by 60%, to $19.0 \pm 1.7$ m/yr in 2015–2016 (Fig. 2a). After 2016, GL basal melt rates remained at high rates >$17.0 \pm 1.5$ m/yr averaged across the ice shelf width. A similar evolution is observed for Ryder: basal melt decreased from $47.6 \pm 4.2$ m/yr in 2002–2003 to $38.4 \pm 3.6$ m/yr in 2004–2008. This was followed by an increase in GL melting rates to $48.0 \pm 3.5$ m/yr in 2014, or 25% higher (Fig. 2c). While the largest increase in melting rates are usually observed close to the grounding line, where the ice shelf draft is maximum (Fig. 2a–d), we observe for Ryder that the largest variability is measured at draft values of 300–400 m. In this area, basal melting increased from near no melting up to 25 m/yr between 2002 and 2020 (Fig. 2c). This also suggests that the GL is isolated from the largest increase in the water column temperature. After 2014, GL melt rates remained constant, at an average value of $47.0 \pm 3.1$ m/yr. For Steensby, the melt rates increased from $7.0 \pm 4.5$ m/yr in 2002–2007 to

a peak of $19.7 \pm 4.6$ m/yr in 2014 during the breakup. Basal melting has then remained constant at $12.5 \pm 2.4$ m/yr in 2016–2020, or almost twice as large as the melt rate for 2002. GL melt rates measured at 79 N averaged $21.1 \pm 2.1$ m/yr in 2006–2011 and increased by 37% up to $29.0 \pm 2.4$ m/yr in 2020 (Fig. 2). Spatially, the largest melt rates values are found in the center of the GL, except for Ryder, where melt rates are higher in the eastern section of the GL. Maximum GL melt rates are found for 79 N, where it often exeeds 80 m/yr, and can reach up to more than 100 m/yr in agreement with in-situ measurements[19]. Over the entire period we did not detect significant changes in melt rates for SB, which averages $4.6 \pm 7.0$ m/yr between 2002 and 2018 (Fig. S6).

Overall, the volume of ice shelves in North Greenland decreased from $957.0 \pm 8.5$ km$^3$ in 2000 to $704.0 \pm 7.8$ km$^3$ in 2012, equivalent to a loss of 26%. Between 2013 and 2022, the total volume stabilized and slightly increased to $750 \pm 7.7$ km$^3$, because of widespread GL retreat and the frontal advance of Petermann, which increased the ice shelf area. Using an historical DEM[20], we calculate a total volume of $1149.5 \pm 55.1$ km$^3$ in 1978. We conclude that ice shelves of North Greenland have lost 35% of their volume during the last 45 years (Fig. S16). Similarly, the total area of floating ice dropped from 5386.6 km$^2$ in 1978, down to 3305.8 km$^2$ in 2013–2022, hence losing more than one-third of its original extent (Fig. S16).

### Partitioning of ice shelf mass losses

Overall in the period 2001-2021 (Fig. 3), ice shelves mass losses due to basal melting total $331.3 \pm 52.8$ Gt vs $222.8 \pm 55.8$ Gt from calving fluxes and $38.4 \pm 6.5$ Gt from SMB (Fig. 3). Specifically, over this period, mass losses due to basal melting dominate over increased calving with $152.2 \pm 27.0$ Gt (melt) vs $73.0 \pm 29.0$ Gt (calving) for Petermann, and $46.2 \pm 10.0$ Gt vs $4.0 \pm 12.0$ Gt for Ryder (Fig. S26). For Steensby, mass losses from calving overwhelm basal melting, with a total loss of $2.7 \pm 1.7$ Gt from basal melting and $6.6 \pm 2.3$ Gt from calving (Fig. S28). This is mainly due to the large breakups that occurred between 2012 and 2014 (Fig. S3). For the case of 79 N, basal melting totals $126.7 \pm 43.0$ Gt and $184.0 \pm 45.4$ Gt from calving. We note that for this glacier, the share of mass losses owing to basal melting has increased after 2012 from 35% to 39% in 2021 (Fig. S28). For SB, in the period 2001–2013, the calving totals $7.4 \pm 5.0$ Gt/yr against $2.7 \pm 2.8$ Gt/yr for basal melting. After 2013, the ice shelf gained mass, which is mainly attributed to the grounding line retreat that occured in this time period, rather than a real ice shelf frontal advance (which remained stable) and which is a limitation of our approach for the specific case of SB (see Methods).

### Glacier dynamical response

Ice shelf changes were followed by important glacier dynamical responses. After the partial collapse of Steensby in 2014, the GL velocity increased by more than 60% to $451.5 \pm 43.0$ m/yr in 2020 (Fig. S12). Similarly, the 2012 calving event of Petermann was followed by a speed increase of 10–15%[18]. Our dense time series shows that the surface flow velocity of Ryder increased from $467.6 \pm 17.0$ m/yr in 2000–2013 to $590.1 \pm 49.1$ m/yr in 2018 (or 26%) before slowing down to $543.5 \pm 39.0$ m/yr in 2020 (Fig. S13). Finally, the ice velocity of 79 N consistently increased from $1500.0 \pm 100.0$ m/yr in 2000 up to more than $2100.0 \pm 41.0$ m/yr in 2020, or by 40% (Fig. S14). For Zachariæ Isstrøm, we expand on previous studies[10] and show that the glacier continued to accelerate from 1200 m/yr in 2000 to 2900 m/yr in 2019 (Fig. S10). No changes in ice dynamics are observed for the other glaciers (Figs. S9–S15).

These dynamical changes are also reflected in the yearly discharge estimates and are consistent with the observed evolution of basal melt close to the GL (Fig. 3). The discharge of Steensby increased by 28% between 2000 and present. Discharge rates continued to rise while basal melting stabilized after 2015 (Fig. 3). For Petermann, the ice

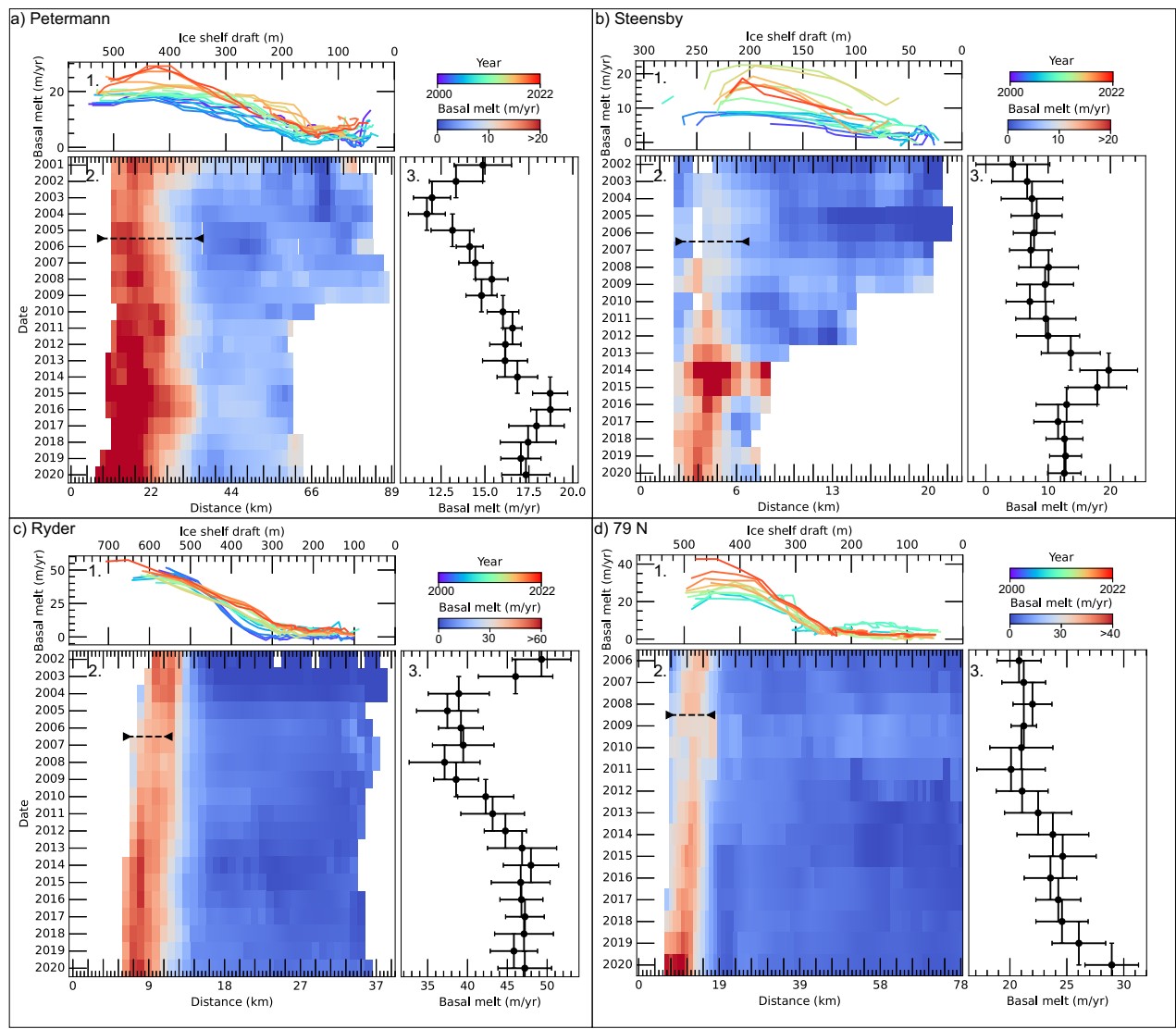

**Fig. 2 | Evolution of basal melting rates over four ice shelves.** Melt rate evolution is provided as a function of the ice shelf draft averaged across the ice shelf width all along the centerline (**a–d panels 1**), Fig. S8. Change in basal melt rate is represented as an hovmöller diagram along the ice shelf length (**a–d panels 2**), where basal melt rates are averaged across width. Grounding line melt rate averaged inside the black dotted line for all years is also provided, with error represented as vertical bars (**a–d panels 3**). The region of averaging was chosen on a case by case basis to focus on the grounding line region, where melt rates are the highest. Note the change in basal melt scale bar for each panel.

discharge started to increase in 2010, and reached a maximum in 2018 at $11.7 \pm 1.2$ Gt/yr, two years later from the peak in basal melt (Figs. 2, 3). Interestingly, for Petermann and Ryder, the slowdown in ice discharge observed after 2018 is coincident with stabilized GL basal melt (Fig. 3). The GL discharge of 79 N increased by 14% from $11.6 \pm 0.8$ Gt/yr in 2000 to $13.2 \pm 0.7$ Gt/yr in 2022 (Fig. 3). These results suggests the strong control of basal melt rates on glacier dynamics.

### Changes in ocean conditions

The Norwegian Atlantic Current advects warmer water northward. A branch of this current flows east of Svalbard across Fram Strait and directly toward NEG[21]. In contrast, the waters in WNG (Fig. S17) are branching from the Arctic Transpolar current down south into Naires Straits and along the North coast of Greenland. Analysis of CTD and AOPR highlights different thermal regimes across the entire North Greenlandic region. In WNG, we note that the ocean temperature at depth (250-450 m), modestly increased by 0.1 °C in the period 1965–2000, from −0.1 °C to 0.0 °C (Fig. S17). Between 2000 and 2015, we found a larger increase in temperature, from 0.0 °C to 0.25 °C. In

NEG (Fig. 3), CTD measurements show that the average temperature at depth is 0.8 °C higher than WNG (Fig. 3). In this region, the increase in water temperature was more important and started earlier than WNG: between 1980 and 1990 the temperature increased from 0.1 °C to 0.4 °C (Fig. S17d). In the period 1990–2020, the change was more than twice larger and increased from 0.4 °C to 1.2 °C. For NEG, we observe a high peak in water temperature in 2010, with a magnitude similar to the one reached in 2020. For WNG, the highest temperature peak was reached in 2015, and ocean temperature decreased since then down to 0.1 °C (Fig. S17).

## Discussion

While we measure large changes in glaciers and ice shelves, the timing of events are heterogeneous. The earlier observed ice shelf collapse was recorded at Ostenfeld in 2003. CTD and model reanalysis only show a modest increase in ocean temperature in that sector (Fig. S17). The analysis of optical imagery shows that the ice shelf had no lateral contact with the fjord margins since 1978 (Figs. 1, S1). We also note the absence of ice mélange after 1978–1992, which may have buttressed

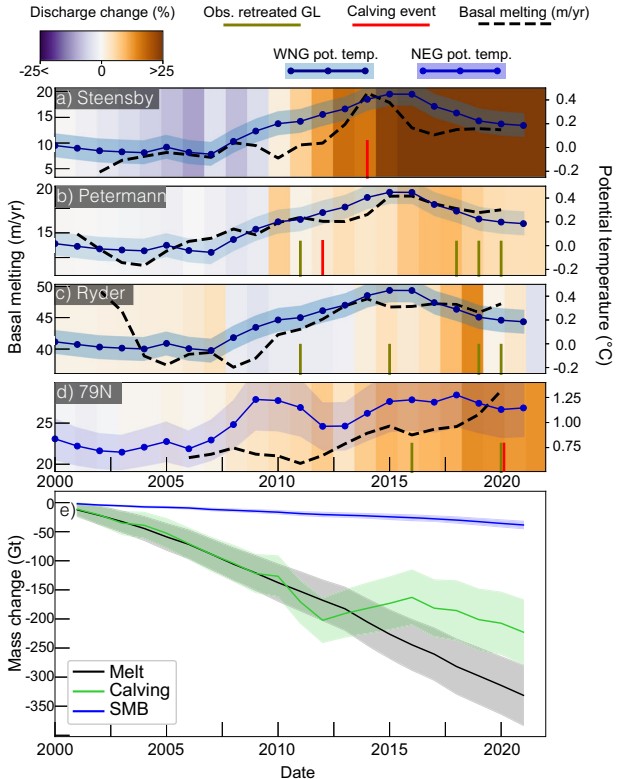

**Fig. 3 | Glacier dynamical response to ice shelves weakening. a–d** Changes in ice discharge is represented in % relative to the average of 1970–2000, and color-coded from purple to brown. Ice shelf melt rate evolution is represented as black dashed line. Calving event and observed dates of GL retreat are noted as vertical red and green bars respectively. Changes in ocean temperature between 200 m and 450 m is plotted as blue solid line for North East Greenland (NEG) and Western North Greenland (WNG) regions. Error on potential temperature is plotted as a shaded area. The area where potential temperatures are calculated are shown in Fig. S15. **e** Cumulative mass loss changes owing to basal melting, calving and SMB for the five remaining ice shelves (cf Fig. 1), with errors plotted as a shaded area.

the ice shelf prior to collapse[22] (Fig. S1). The absence of dynamical response (Fig. S9) further confirms the minor role of Ostenfeld in providing buttressing to the glacier. The collapse of Hagen Bræ corresponds with a previously reported surge in 2007–2015[23] (Fig. S11). In the southern section of the shelf we measure large basal melting rates, which combined with the important strain rates, have made the floating section prone to collapse (Figs. S4, S5, S7). Earlier observations of changes in ice thicknesses and basal melting in the 1990–2000 would be needed to further detail the processes and exact timing of events that have led to the collapse of these ice shelves.

For the remaining ice shelves, basal melting rivals the highest rates observed in the Amundsen sea Embayment of Antarctica[24–26]. The observed increase in melting coincides with a distinct rise in ocean potential temperature, suggesting a strong oceanic control on ice shelves changes. Calculated correlation coefficients between basal melt rates and ocean temperatures exceed 0.9 for Petermann, Ryder and Steensby, and 0.5 for 79 N (see Methods). The general increase in water temperature for the period 2005–2016 corresponds to the calculated rise in basal melting at the GL of Petermann, Steensby and Ryder (Fig. 3a–c). Peaks in melt observed in 2014–2016 at Petermann, Steensby and Ryder are consistent with the highest ocean temperature in that sector during the same period (Fig. 3a, c). Similarly, constant or decreasing melt rates in the period 2000–2005 and 2016–2019 match periods of decreasing ocean temperatures (Fig. 3b, c). While ocean thermal forcing has slightly decreased in WNG, it continued to rise at

the front of the 79 N, with consistently increasing melt rates in 2006-2021 (Fig. 3). Overall, the warmest ocean temperatures are observed in NEG (>1 °C in 2020, Fig. 3d) and correspond to the maximum melt rates values, which are observed at 79 N (Figs. 3, S8 and S17). We note that the largest GL retreats are observed where basal melt rates are the highest, i.e., the central section of Petermann and 79 N's GL, and the eastern side of Ryder's GL (Fig. S8). For Steensby the large calving event of 2014 is consistent with the highest measured basal melt value of the entire time series (Fig. 3).

Regional atmospheric climate model outputs show a similar trend in the surface runoff for all drainage basins. The runoff increased between 1990 and 2010, after which it stabilized at rates of 4.5 Gt/yr, 1.5 Gt/yr and 2.8 Gt/yr for Petermann, Steensby and Ryder respectively, with a large inter-annual variability after 2010 (Fig. S26). For 79 N, runoff increased from 2.1 Gt/yr in 2000 and stabilized at 5 Gt/yr around 2007 (Figs. S26, S27). After the runoff stabilization, basal melting rates continued to increase, hence suggesting that runoff played a minor role in its evolution (Figs. S26, S27). Using the parametrization from[26] over Petermann and neglecting the role of subglacial water discharge, we show a good agreement between modeled and observed GL melt rates (Fig. S29), which further supports the interpretation that changes in runoff had a negligible influence on basal melt rates.

In the case of Ryder, Petermann and 79 N, increased basal melting is accompanied by GL retreat and followed by an increase in ice discharge (Fig. 3b–c). The lack of grounding line data, specifically during the period of 1997–2010 when basal melting significantly increased, is a limiting factor in accurately assessing the precise timing relationships between these two processes. The comparison of the spatial variability of basal melting with shear strain rates and crevasse formations over Petermann shows a close spatial correlation between subglacial melt channels and recently formed fractures in 2015[18] (Fig. S30). This suggests basal melting may be playing a complex and crucial role in thinning the ice shelf from below, and modulating the GL position and glacier dynamics, hence making it prone to enhanced fracturing. For Steensby, the changes in ice discharge observed after 2014 suggests that the glacier responded to a loss in ice shelf buttressing, with a strong interplay between enhanced basal melt rates and the large calving event.

Subglacial bedrock topography can also exert a strong control on the retreat rate of glaciers[12]. Currently, the eastern section of the GL of Ryder stabilized on a prograde bedslope grounded at 700 m below sea level (Fig. S18). The western part is however sitting on top of a deep retrograde bed at −400 m, which deepens over the next 6 km to −740 m. For 79 N, the central part of the GL is at −520 m and sitting on top of a downsloping bedrock that goes down to −640 m over a distance of 4 km (Fig. S18). A similar setting has recently been reported for Petermann[18], which could face a retreat of another 8 km before the GL stabilizes (Fig. S18).

Continued ocean and satellite observations are key to provide insights on how these ice shelves will respond to future climate forcing. High resolution ocean models and bathymetry mapping should be used, together with CTD deployments, to provide insights into warm water intrusions in fjords and ice shelves cavities[27]. Basal melting is a complex process, and one of the main sources of uncertainties in future projections of the ice sheets contribution to SLR[3,28]. We provide observations of basal melt rate at an unprecedented level of resolution which opens the door to a higher degree of understanding of ice shelf processes. This allows reanalysis data to validate coupled ice-ocean models and to better estimate parametrization of ice-ocean interaction processes. This will ultimately provide insight into the future of these glaciers as well as the fate of larger ice shelves in Antarctica[28].

Our results document a holistic overview of glacier-climate-ocean interaction in North Greenland. We are able to identify a widespread ongoing phase of weakening for the last remaining ice shelves of this sector. The GL are exposed to the warmest water layers and currently

sitting on retrograde bed slopes. This makes them extremely vulnerable to unstable retreat and ice shelf collapse if ocean thermal forcing continues to rise, which is likely to be the case in the coming century[29,30]. A loss in the buttressing provided by ice shelves in this sector will likely trigger an increase in the discharge[13,14] that could rival the largest contributors to Greenland ice mass loss. This could have dramatic consequences in terms of SLR, as it is the sector in Greenland with the greatest SLR potential (2.1 m)[3,4].

## Methods

### Ice surface elevation and volume change

In this study we make an extensive use of all available surface elevation data to reconstruct a comprehensive yearly history of ice shelf thickness changes and basal melt rates between 2000 and present. Airborne altimetry measurements from NASA's Operation Icebridge Airborne Topographic Mapper (ATM) and LVIS (Land, Vegetation and Ice Sensors) were used to document ice shelf thickness changes from 1993 to 2016[31,32]. Satellite altimetry from NASA's Ice, Cloud and land Elevation Satellite missions (ICESat-1) were also used to document the evolution of ice shelf thickness between 2003 and 2009, after which the satellite was retired due to a laser failure. Between 2018 and present, we used satellite altimetry measurements from the recent ICESat-2 mission. Quarterly digital Elevation models (i.e temporally averaged) derived using Digital Globe imagery as part of the Greenland Ice Mapping Project were used between 2012 and 2016[33,34]. We also used DEMs derived from Synthetic Aperture Radar using NASA's GLacier and Land Ice Surface Topography Interferometer airborne (GLISTIN-A) which measured surface elevations in 2016-2019 around the periphery of the Greenland Ice Sheet using Ka-Band (8.4 mm wavelength) single-pass interferometry[35]. Note that the GIMP and GLISTIN-A DEM are co-registered to airborne altimetry data (see below).

DEMs from ASTER imagery are generated at 30 m resolution between 2000 and present with the MMASTER processing chain, which is based on the MicMac photogrammetry software[36]. These observations are aligned horizontally and vertically following the classical scheme from[37], and using yearly mosaics of satellite altimeters as ground control points on land and ice[38] (see Fig. S39 for examples). The reference datasets were assembled using all altimetry data on stable ground (without ice) and elevation measurements from the same year as the DEMs on grounded and floating ice[24]. When all ASTER DEMs are generated and coregistered, we filter the elevation map with respect to correlation score (ranging between 0–100)[36]. We typically remove all pixel with correlation score below 85, and stack all elevation maps within a year into a composite mosaic. After stacking, the resulting DEM is again aligned horizontally and vertically to the relative reference surface elevation. Before stacking, we find an average standard deviation of the difference with satellite altimetry that is ranging between 6 and 9 m. We calculate the root mean square error between all ASTER DEMs and the latest version of the GIMP digital elevation model[39]. We removed outlier pixels with differences exceeding 200 m compared to the latest version of the GIMP v2[34]. For elevations above 1000 m (the accumulation area), we additionally filtered pixels with differences over 75 m from the GIMP v2 DEM, considering the anticipated lower dh/dt values in this region. These threshold values were chosen arbitrarily through multiple rounds of iterative filtering tests.

For the GIMP, GLISTIN and ASTER DEM, we additionally calculate the yearly difference with the corresponding altimetry data on the ice shelf: if the average difference exceeds 1 m, the DEMs are shifted vertically, in order to be centered on zero. Overall, we find a mean standard deviation of the difference on ice shelves of 2 m on ASTER DEM (after stacking) and GLISTIN-A, and 0.8 m for GIMP (Figs. S31, S38). For Petermann, Ryder and Steensby, ASTER DEMs from 2000-2001 didn't have altimetry data on the ice for coregistration, hence we used the closest reference DEM in time to align DEMs. Uncertainties on the calculated melt rates might therefore be higher for these dates (see

Fig. 2). For 79 N, we were not able to find ASTER DEM with a satisfying signal to noise ratio between 2000 and 2006.

Finally, mosaics of surface elevation for each ice shelf are assembled by merging all available elevation data from all available sensors, and by assigning higher priorities to the best vertical accuracy (e.g, altimetry). This ensures to have the most comprehensive spatial coverage and highest vertical accuracy on all ice shelves around Greenland. Ice shelf volume changes are calculated by converting the surface elevation of the ice shelf into ice thickness using the hydrostatic equilibrium equation within an ice shelf mask, with an ice density of $0.917$ g/cm$^3$,[40],and a water density of $1.028$ g/cm$^3$. Grounding line and ice front mapping determination is described below. Uncertainties in the ice volume are calculated by assuming an error in ice shelf area of 1 pixel at the ice front[41], and the corresponding error in ice thickness using the error on the surface elevation (see above). We also consider an additional uncertainty of 1 m, for changes in thickness due to firn air content, as described in ref. 40. Mean firn air content in this region is typically <1 m, with change rates of less than 1 cm/yr[33].

### Grounding line and ice front mapping

We use Interferometric Synthetic Aperture Radar (InSAR) data from ESA's Earth Remote Sensing radar satellite (ERS-1) acquired in 1992 with a 3-day revisit time, in 1995/1996 from the ERS-1/2 tandem mission, and in 2011 from a 3-day revisit time before the end of ERS-2 mission. Data are downloaded via the ESA Online Dissemination Service as single look complex (SLC) scenes and processed using the GAMMA Software[42]. We measure the tide-induced vertical motion of ice using a Quadruple Differential SAR Interferometry approach (QDInSAR)[43]. Phase coherence is maintained in fast flowing regions by coregistering SLC data using speckle tracking[44,45]. We calculate interferograms with the phase difference between the co-registered SLCs. The grounding line position is obtained by differentiating two interferograms spanning the same time interval, after correcting for topography[43]. We use the GIMP v1 DEM time-tagged in 2007 to remove the topographic signal[34], assuming that no changes in surface ice elevation has occurred in the time interval between the DEM and SAR data. For 2014–2021, we use Sentinel-1 (S1) data with a repeat cycle of 6 to 12 days. Phase jumps at burst boundaries were accounted for using the TOPS coregistration method[44]. We use the GIMP DEM v2 time-tagged in 2014 to correct for the topographic phase for S1[34]. We map the inward limit of detection of vertical motion, where the glacier first lifts off its bed[43]. We processed around 1000 interferograms for Petermann, 600 for Ryder, 520 for Steensby and 500 for 79 N, which provide us with a good idea of the grounding zone (>80% is from Sentinel-1 imagery). We measure the distances of maximum grounding line retreat or advance relative to the earliest most retreated grounding line observation date. Example of grounding lines and interferograms are provided in Fig. S39.

We manually digitized yearly ice shelf frontal positions between 1990 and 2021 using summer imagery from NASA's Landsat-1-8 optical satellite, ESA's Sentinel-1/2 satellite. We used the Google Earth Engine Digitisation Tool (GEEDiT v1.012) developed by James M Lea at the University of Liverpool[46].

### Ice velocity

We monitor the evolution of the glacier dynamic state velocity by calculating the surface displacement from three different satellite sensors from images collected between 2013 and 2021. Two of them, ESA's Sentinel-2 (S2) and NASA's Landsat-8 (L8), are optical imagers and one, ESA's Sentinel-1 (S1), is a synthetic aperture radar operating in C-band. We use persistent surface features or speckle to map ice displacements between two consecutive images. We calculate the normalized cross-correlations between the reference and search image chips using repeat cycles shorter than 30 days for Landsat-7/8 and Sentinel-2, and 12 days for Sentinel-1[47,48]. Between 1999 and 2012 we

supplemented our Landsat-7 ice velocity record with repeat cycles ranging from 336 to 400 days. For L8, S2, and S1, sub-images of 32 × 32, 32 × 32, and 192 × 48 pixels are used, respectively. We calibrate our displacement maps by taking advantage of the ice velocity products from prior surveys in[47]. The final calibrated maps are resampled to 150 m posting in the north polar stereographic projection (EPSG:3413). The time series established is completed by historical measurements made from ERS-1/2, RADARSAT-1, ALOS/PALSAR, ENVISAT/ASAR, Landsat 4 to 7 and TerraSAR-X[4,33,47–50].

In order to monitor changes in rates of ice deformation, we derive the evolution of the shear strain rate for 2000 and 2019 using annual ice velocity mosaics[2]. The annual mosaics are assembled from the same observations as described earlier, using a strict procedure of spatial and temporal filtering. These data provides the most accurate ice velocities values with the best signal to noise ratio and lowest uncertainties in ice flow direction[47]. Strain rates were retrieved using the same methodology as described in[18,51].

## Ice shelf basal melt rates

Changes in ice shelf thickness can be caused by (1) the rapid advection of ice, (2) surface mass balance, (3) firn air content, and (4) basal melting, can be summarized through Eq. (1):

$$\frac{Dh}{Dt} = \frac{(\rho_w - \rho_i)}{\rho_w}\left(\frac{M_s}{\rho_i} - H_i \nabla.v - w_b\right) + \frac{Dh_{air}}{Dt} \qquad (1)$$

Where $\frac{Dh}{Dt}$ is the total Lagrangian thickness change, H the ice thickness, $\nabla.v$ the velocity divergence, $M_s$ the surface mass balance, $\frac{Dh_{air}}{Dt}$ the change in firn air content and $w_b$ the basal melt rate. The ice density is taken as 917 kg m$^{-3}$ and the ocean water density as 1.028 kg m$^{-3}$.

Changes in ice thicknesses are determined using a Lagrangian framework, i.e., we track the evolution of every pixel in a given DEM to its downstream location in another DEM acquired later in time. We follow the work of ref. 38 and calculate the trajectory of every pixel along flow paths calculated from yearly surface flow velocity fields, (time step of 15 days), to appropriately account for changes in ice dynamics[38]. To avoid artifacts in ice velocity mosaic we smooth the ice flow observation using a 2.5 km rolling median filter[38]. Because the shear margin of glaciers is a region where surface flow velocity errors are the highest, we manually filter glacier boundaries to avoid artefacts in the thickness changes and ice flow divergence derivation (see below).

Changes in ice volume Dh/Dt are determined using yearly DEMs described above and covering the years 2000–2020. The ice shelf area dynamically evolves every year with changing ice front and grounding lines. We calculate a dense time series of basal melt rates using all possible combinations of DEMs since 2000, with time difference between the DEM sources spanning from 2 year (low Signal to Noise Ratio-SNR) to 6 years (best SNR). Several tests were conducted with DEMs separated by 1 year, but the SNR was insufficient for use in long-term melt rate trend interpretations. Prior to the calculation of the Dh/Dt term, we corrected every DEM over vertically induced tidal motion using the pyTMD toolbox, which reads outputs from the AOTIM-5 inverse tide model[52]. To account for changes in thicknesses due to surface melting, we use the monthly version of the Modele Atmospheric Regional (MAR) at a resolution of 1 km[53]. Changes in surface mass balance are also calculated in a Lagrangian Framework using annual values of SMB. Mean firn air content in this region is negligible, hence we did not account for it in the melt rate calculation[40] (see "Ice surface elevation and volume change section").

We calculate changes in ice thickness due to the advection using the ice flow divergence based on yearly maps of surface flow velocity as described previously. Finally basal melt rate maps are smoothed using a rolling median of 450 m. For each pair of DEMs, we use the central date to reference the basal melt map. For all ice shelves, we stacked all available melt maps within a year to produce most accurate time series of basal melting. The evolution of the melt history is plotted using an hovmöller diagram (Fig. 2), which is the width-averaged basal melting along an ice shelf flowline (Fig. S8). Uncertainties in basal melting rates are calculated using standard error propagation methods and assuming a conservative error of 15% on the SMB[54]. This was calculated considering Fig. 5 from ref. 54 and the SMB values <−3 mWE which are the ranges with the largest discrepancies between observations and regional climate models. An arbitrary and conservative error of 100% is applied on the ice flow divergence. We use the related yearly errors on the DEMs that are used for calculating the changes in ice thickness (see Section "Ice surface elevation and volume change")

## Grounding line ice discharge

Due to the likely high uncertainties in bedrock elevation at the grounding line[7,12], that can reach hundreds of meters, we decided to use estimates from[4] for the period 1990–2017 and an updated version of the calculation of[8]. For 2017–2022, we evaluate changes in ice discharge based on the PROMICE solid ice discharge data product, calculating ice flux through gates located approximately 5 km upstream of the grounding line (Fig. S18). The flux gates discharge obtained are generally higher than the estimate of[4] as these are not corrected for the surface mass balance downstream of the flux gate. We therefore scale our new discharge estimates, based on the mean bias on the overlapping periods of measurements, so that they match the median values of ref. 4.

## Calving fluxes

Yearly calving fluxes were calculated for the still-standing ice shelves (Steensby, Petermann, Ryder, 79 N, Bistrup/Storstrømmen) using an input-output approach[55]. We estimated this flux over the entire studied period using the calculated mass changes from the ice shelf volume variations, basal melt rates, grounding line discharge and changes in surface mass balance. Positive calving flux values are typically found when the ice shelf area increased, due to a frontal advance or large grounding line retreat. The drawback of this methodology lies in the case of positive calving values, as it doesn't allow for discrimination between an increase in volume due to retreat of the grounding line or advancement of the front. For missing values of basal melt rates, time series were extrapolated to obtain a comprehensive time series over the period 2001–2021. The melt rate value of 2002 was used for 2001 in the case of the Steensby ice shelf and we used the 2005 basal melt values for 79 N over the period 2002-2005, which might bias the mass losses owing to basal melting. Uncertainties in calving fluxes are calculated with the square root of the sum of squared errors on the grounding line discharge, the ice volume, and the basal melt rates. Due to the quiescent phase of Bistrup/Storstrømmen, the grounding line discharge is often less than 0.5 Gt/yr over the entire study period. Futhermore, the calculated basal melt rates are <10 m/yr, and we only observe moderate changes in SMB, which averages −0.25 Gt/yr in 2019–2021 over the shelf. These changes are moderate, or non-existing, and our approach does not allow to highlight a clear environmental trigger on the large increase in ice mass attributed to the grounding line retreat observed after 2013.

## Ocean conditions

In order to monitor the evolution of ocean thermal forcing, we use conductivity temperature and depth measurements from the Hadley centre (bodc.ac.uk) spanning 1960 and 2019, combined with CTD from NASA's Ocean melting Greenland campaign from 2016 to 2021[17]. In addition to the in-situ observation, we also document the spatial variation of ocean thermal forcing using the Arctic ocean physics

reanalysis (AOPR, https://doi.org/10.48670/moi-00007) produced by the Nansen Environmental and Remote Sensing Center, Norway, and distributed by the Copernicus Marine Environment Monitoring Service. The AOPR captures the trends in the CTD in-situ measurements and allows us to interpret changes in ocean temperature with a finer temporal scale. The ocean and sea-ice model assimilate CTD profiles as well as remotely sensed data such as sea surface temperature, sea-ice concentration and sea surface height, for the North Atlantic Ocean and Arctic between 1992 and 2020[56]. The reanalysis is provided on a horizontal grid cell size of 12.5 km × 12.5 km and 40 vertical levels, and monthly outputs are used to calculate yearly ocean thermal forcing (potential temperature minus freezing temperature). In North Greenland, grounding line depth, sub-ice shelf, and fjord bathymetry are highly uncertain and results from rough interpolation (Fig. S18). Furthermore, the availability of CTD measurements in the fjords is sparse, and specially over the study period (Fig. S18). In order to maximize our confidence in the trends of ocean temperature, we decided to analyze regional values for ocean potential temperature in the western North Greenland and north east Greenland (see Fig. S18). For each box (Fig. S17c), we extract average temperature below depth 200 m and 550 m. These temperature ranges were chosen with respect to CTD profiles for each region (Fig. S17). Within each region we evaluated the bias between in-situ data and reanalysis model and found a significant mean bias of 1.1 °C for NEG. We adjusted the depth averaged temperature value accordingly and found afterwards average differences with CTD measurements of 0.06 ± 0.02 °C for WNG and 0.05 ± 0.28 °C for NEG (Fig. 3). We use these statistics as uncertainty measurements in the reanalysis temperature trend in Fig. 3. We use the temporal evolution of ocean temperatures to compare them to the evolution of basal melt between 2000 and 2021. Given the temporal and spatial uncertainty of the datasets, we decided to smooth the time series with a moving average of 4 years (the average baseline used for melt rates). In this way the correlation coefficient is calculated on general trends over the whole study period. Indeed, uncertainties can be caused by the temporal resolution of our basal melting time series, which averages different temporal baselines, and the large spatial averaging on the AOPR. Additionally, AOPR may not fully capture the ocean dynamic and specifically in 2000-2015, where few in-situ measurements exist (Fig. S17).

## Surface mass balance
We investigate changes in surface mass balance using a simulation from the Modèle Atmosphérique Régional (MAR-v3.12[53]) forced by the ERA5 reanalysis[57]. The simulations were run over Greenland at a resolution of 11 km, then statistically downscaled at a resolution of 1 km.

## Reporting summary
Further information on research design is available in the Nature Portfolio Reporting Summary linked to this article.

## Data availability
The grounding line, ice front positions, surface flow velocity, basal melt and calving rates have been deposited in the Zenodo database and can be accessed at https://doi.org/10.5281/zenodo.8354794. Outputs from the Modèle Atmosphérique Régional are available at http://phypc15.geo.ulg.ac.be/fettweis/MARv3.12/Greenland/. This study has been conducted using E.U. Copernicus Marine Service Information[58]; https://doi.org/10.48670/moi-00007. CTD measurements are freely available at https://climatedataguide.ucar.edu and https://podaac-tools.jpl.nasa.gov/.

## Code availability
Codes used to produce the figures of this paper can be accessed at https://doi.org/10.5281/zenodo.8354794.

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

## Acknowledgements

This work was supported by the French National Research Agency, grant no. ANR-19-CE01-0011-01 (R.M, E.J, J.M), and grant no. and ANR-19-CE01-0015 (P.M). This work was also supported by the Villum Young Investigator grant no. 29456 (R.M, A.B). S.H.L. was funded by the PROMICE project (www.promice.org). N.C.J. was funded by EU-H2020 grant no 101003536 (ESM2025). M.H.W. was supported by awards from the NASA Cryospheric Sciences Program (NNH20ZDA001N-CRYO) and the NASA Physical Oceanography Program (NNH22ZDA001N-PO). This work is dedicated to Jeremie Mouginot, leader of the ANR SOSice project, and who tragically passed away in September 2022.

## Author contributions

R.M., J.M. designed and conducted the study. All authors (E.J., M.H.W., J.M., S.H.L., P.M., N.C.J., A.B.) contributed to writing the article and interpreting the results.

## Competing interests

The authors declare no competing interests.
