## [Peer Review File · Nature Communications]

Rapid Disintegration and Weakening of Ice Shelves in North GreenlandReviewers' Comments:

Reviewer #1:

Remarks to the Author:

Summary

The manuscript describes changes in the extent and volume of northern Greenland ice shelves. The analysis focuses in particular on grounding line retreat and its relationships with basal melting and calving as well as ice discharge. The authors observe moderate to strong correlations between basal melt rates and ocean temperatures at several glaciers and conclude that the ice sheet's remaining ice shelves will likely continue to melt at accelerated rates and potentially destabilize in the coming decades.

The authors appear to have addressed feedback from previous reviewers, resulting in a good description of a novel dataset and well-supported conclusions regarding shelf stability. My only comments are in regard to wording in a few locations and adjustments to color ramps in figures.

Major Comments

1. I'm a bit confused by the beginning of the summary of shelf change for the non-collapsed shelves. On line 66 you state that the grounding line of Steensby was stable since 1992. Then in the next sentence you state "In August 2014, after enhanced fracturing, this ice shelf shrank...". After carefully rereading these sentences it seems as though the terminus retreated but that had no appreciable influence on the grounding line. If this interpretation is correct, I recommend that you slightly revise those introductory sentences to make it no grounding line retreat occurred despite terminus retreat. Possibly revise the end of the first of these two sentences to simply to say "despite considerable terminus retreat" or something similar.
2. I strongly advise changing some of the color ramps used in Fig. 1 because it is difficult to discern some of the terminus and grounding line positions where they overlap the velocity map due to similarities in color. Some of the Steensby terminus positions and grounding lines for Petermann and 79N are very difficult to see.

Similarly, the same blue-white-red color ramp is used in Fig. 2 to show spatio-temporal variations in basal melt rate and in Fig. 3 to show discharge change. If possible, use unique color ramps to represent different metrics.

Line-by-line Comments

- line 33: Change "remain" to "remained"
- lines 39-42: I recommend switching the order of these two sentences so that the one describing the volume ("Overall ...") comes before the one stating it is important to study these sites ("It is thus...").
- line 87: Change to "While the largest"
- line 93: Replace "twice the larger than in 2002" with "twice the melt rate for 2002"
- line 113: Replace "happening" with "happened"
- line 156: The meaning of "is coherent with" is unclear. Replace with "is consistent with" or "is coincident with" depending on your meaning.
- line 174: Remove "however"
- line 189: Replace "coefficient" with "coefficients"

Reviewer #2:

Remarks to the Author:

This is paper tells a compelling story of ocean heat increasing basal melt, which in turn impacts ice shelf stability and the discharge of ice draining into the ice shelves. The authors did an enormous amount of work pulling together a diverse dataset and creating clean and illustrative figures.

I have two main concerns with this paper, which prevent me from recommending publication at this time. First, in many places the methodology is lacking detail. Even in the supplemental, I had questions about how some of their work was completed. These are outlined in my detailed comments

below. Second, I found that in several places, the paper made big conclusions with intermittent evidence, and selective support from the literature. Ice-ocean interactions are complicated, and a lot of interconnected processes occur in concert with each other. One repeated concern that I have in this paper is the tendency to over-state previous findings and their own results in the quest to find a clean and compelling story. By rewording the text in several places, the authors should acknowledge this complexity instead of simplifying it.

detailed comments:

10: capitalize University

13: This might seem totally pedantic, but ice shelves don't "drain" the ice sheet. The outlet glaciers do the "draining" – the ice shelves are the floating extensions of the ice shelves. Either reword that part of the sentence, or just delete that part of the sentence.

14: capitalize Ice Sheet

16: substantial and widespread increase in mass loss – Based on Figure 3, I'd call this moderate increase in mass loss. Steensby and 79N are the only really convincing ones.

25: Find original reference for this – IPCC synthesizes studies.

27-29: This is overstating the results of your 2018 paper, which focuses on glaciers in the SE, not "around Greenland"

31-33: These last two sentences could use some reorganization – they seem contradictory and sloppily written.

36: which glaciers?

36: None of those references actually calculate backstress at the grounding line. Instead, maybe compare the glacier basin size and potential SLR contribution.

39-41: I agree with this sentence – that it's important to understand the timing of calving and the glacier response. However, based on everything you've written before this sentence, it sounds like a solved problem. Revisit those references to highlight the complexity and uncertainty more accurately in this cause-effect of calving and acceleration.

41-42: I find this sentence confusing – I think you're referring to glaciers, not ice shelves.

43: Buttressing has never really been quantified, and their role in glacier acceleration is a long-lasting debate in our field. Add references and/or acknowledge that this is a theoretical argument.

46: In other places (e.g. 31) you capitalize "North"

47: Are these SLE values what has already been contributed? Or their potential if glaciers accelerate?

51: datasets

52: Conductivity Temperature Depth

53: ice shelves'

63: the GL "of" Zachariae

65: delete "undocumented"

65: Abstract mentions four remaining ice shelves (16), as does Figure 2.

69: I'm confused by this sequence of sentences that include cumulative retreat and episodic frontal advance. Reword.

70: When switching description to a new glacier, it helps the reader to put the glacier name at the start of the sentence, not the end. "At Ryder Glacier, between 1992 and 2011..."

65-81: In this paragraph, you refer to GL retreat in both area (73) and distance (74, 75, 77, 78). Keep it consistent and maybe add "average" distance to highlight across-width variations.

84: remove ~ if you have uncertainties

84: looks like Petermann melt values could have been higher before 2001, and lower again after 2016. Why compare an average of values over 3 yrs (2001-2004) with peak values in 2016? Same comparison is used for other glaciers too and seems a bit misleading or confusing.

93: twice as large

108: delete "hence compensating for thinning" (makes it sound like a physical compensation, not just a math compensation)

119: not statistically different

119: I don't follow the "owing to calving fluxes" – how does that link to basal melt rates?

123: I like how this is partitioned and clearly described – just add over what time period.

141: Ice shelf changes were followed...

175: Consider rewording "tipping point" – as it is unclear what you mean here and it is a weighted term. What does it mean to reach its tipping point?

179: S1 shows an aerial image of Ostenfeld with mélange, then no mélange. Are there studies that quantify mélange buttressing support to an ice shelf (not just impacting calving rates to glacier termini)?

174-186: I'm a bit lost in this discussion paragraph – what's the main point here? Reorganize and maybe add a concluding sentence to tie it together.

204: Add a bit more detail here – what model and what kind of runoff (surface?)

258: Quarterly? Does that mean 4x a year? I'm confused how quarterly is used in reference to GIMP DEMs between 2012 and 2016 (are those averaged?).

260: SAR DEMs over what time period?

270: Define what you're using for stable ground. What "ice" is considered stable? Was the distribution of the stable terrain relatively even?

271: RMSE over stable terrain?

271: specify what version

275: What do you mean by "coherency"

277: average standard deviation of what?

278: yearly difference of what?

279: you don't need to shift horizontally at all?

282: didn't have

282: why do you need to co-register the DEMs over ice? Isn't there enough stable terrain surrounding these glaciers to perform co-registration on land?

290: Reference Brunt et al (2010) and add some detail for uncertainties associated with assuming hydrostatic equilibrium near the grounding line. Brunt finds that equilibrium is found ~10 km after the grounding line for most ice shelves.

307: GIMP v1 uses elevations from 1978 to 2009. Are you just using this topo correction over stable terrain or the full extent of the region? If for the whole area, won't there be biases from using a range of temporal datasets?

308: Sentinel-1 (S1)

311: Doesn't this change with tides and will therefore have quite a spatial range?

314: define "present"

316: reference for GEEDIT: <https://esurf.copernicus.org/articles/6/551/2018/esurf-6-551-2018.html>

334: Why isn't the same dataset used for shear strain as described in the paragraph above?

358: annual melt rates are reported in your figures above. Why isn't basal melt calculated annually from the annual DEMs?

363: annual

366: delete rapid

366: a third ice velocity product? I'm confused as to why different velocity products are used for each task.

375: Need references and some explanation for these errors of 15% of SMB and 100% for divergence.

379: Citation #5 only applies to Greenlandic glaciers below 66°N where CReSIS radar data is notoriously terrible and subsequently, so are BedMachine bed elevations. The depth radar collected above 76°N is actually much better.

388: The calving flux methodology seems like it might need some revisiting. How "off" are the calving fluxes? How frequently and of what magnitude are the positive fluxes? Pinning all the discrepancy over SMB (presumably that's SMB over the shelf), doesn't seem like it would account for it, as it is likely small. This requires a bit more digging in to.

426: what box?

Figures:

Figure 1:

- Move color bars for grounding line and ice front to top of figure (or at least above the map), so closer to the relevant panels.
- Not sure you need logarithmic ice velocity, since you aren't highlighting slow flow.
- I can't see the flux gates for E-H
- Ice shelf colors correspond....

Figure 2:

- I'm confused by the black dashed line – does it pertain to a specific year, or is it just the length over which basal melt rates are calculated? Is that what goes into the panels on the right? Why different

distances?

- Add "Note change in basal melt scale bar for each panel"
- Add lines, or bold years where major calving events occurred

Figure 3:

- Glacier dynamical response to ice shelf changes (or, if keeping original phrasing, add apostrophes)

References are inconsistent and several are incomplete (#8, #30)

REVIEWER COMMENTS

Reviewer #1 (Remarks to the Author):

Summary

The manuscript describes changes in the extent and volume of northern Greenland ice shelves. The analysis focuses in particular on grounding line retreat and its relationships with basal melting and calving as well as ice discharge. The authors observe moderate to strong correlations between basal melt rates and ocean temperatures at several glaciers and conclude that the ice sheet's remaining ice shelves will likely continue to melt at accelerated rates and potentially destabilize in the coming decades.

The authors appear to have addressed feedback from previous reviewers, resulting in a good description of a novel dataset and well-supported conclusions regarding shelf stability. My only comments are in regard to wording in a few locations and adjustments to color ramps in figures.

Authors: We would like to thank Reviewer #1 for his/her careful and encouraging comments.

Major Comments

1. I'm a bit confused by the beginning of the summary of shelf change for the non-collapsed shelves. On line 66 you state that the grounding line of Steensby was stable since 1992. Then in the next sentence you state "In August 2014, after enhanced fracturing, this ice shelf shrank...". After carefully rereading these sentences it seems as though the terminus retreated but that had no appreciable influence on the grounding line. If this interpretation is correct, I recommend that you slightly revise those introductory sentences to make it no grounding line retreat occurred despite terminus retreat. Possibly revise the end of the first of these two sentences to simply to say "despite considerable terminus retreat" or something similar.

Authors: Agreed. We have clarified this sentence at L91. The sentence now reads as: We observe a GL retreat for all of them, except Steensby, whose position remained stable since 1992 despite considerable terminus retreat.

2. I strongly advise changing some of the color ramps used in Fig. 1 because it is difficult to discern some of the terminus and grounding line positions where they overlap the velocity map due to similarities in color. Some of the Steensby terminus positions and grounding lines for Petermann and 79N are very difficult to see.

Authors: We have revised the Figure 1 accordingly, and now are using linear colormap for the ice velocity, which is solving most of the issues. We have also changed the colormap of the front- and grounding-line positions to avoid any overlaps. Note that to avoid confusion between grounding line and ice front we have used different symbols for both (solid vs dotted-dashed line).

Similarly, the same blue-white-red color ramp is used in Fig. 2 to show spatio-temporal variations in basal melt rate and in Fig. 3 to show discharge change. If possible, use unique color ramps to represent different metrics.

Authors: We have changed the colormap in Figure 3 to avoid confusion. Please note that this paper has been through 3 peer review steps, within each all reviewers have asked for changes in colorbars, sometimes in contradictory ways. Since the figure are aiming to provide a comprehensive overview of what we would like to present, and since we have numerous processes to show, it is extremely complex to find the best compromise in the colormaps. We

have made our best by trying to address comments of Reviewer #1 and #2 here.

Line-by-line Comments

- line 33: Change “remain” to “remained”

Authors: Done.

- lines 39-42: I recommend switching the order of these two sentences so that the one describing the volume (“Overall ...”) comes before the one stating it is important to study these sites (“It is thus...”).

Authors: Agreed. Done.

- line 87: Change to “While the largest”

Authors: Agreed. Done.

- line 93: Replace “twice the larger than in 2002” with “twice the melt rate for 2002”

Authors: Agreed. Done.

- line 113: Replace “happening” with “happened”

Authors: Agreed. Done.

- line 156: The meaning of “is coherent with” is unclear. Replace with “is consistent with” or “is coincident with” depending on your meaning.

Authors: Agreed. We have clarified the sentence with the use of “is coincident with” at L331.

- line 174: Remove “however”

Authors: Agreed. Done.

- line 189: Replace “coefficient” with “coefficients”

Authors: Agreed. Done.

Reviewer #2 (Remarks to the Author):

This paper tells a compelling story of ocean heat increasing basal melt, which in turn impacts ice shelf stability and the discharge of ice draining into the ice shelves. The authors did an enormous amount of work pulling together a diverse dataset and creating clean and illustrative figures.

I have two main concerns with this paper, which prevent me from recommending publication at this time. First, in many places the methodology is lacking detail. Even in the supplemental, I had questions about how some of their work was completed. These are outlined in my detailed comments below. Second, I found that in several places, the paper made big conclusions with intermittent evidence, and selective support from the literature. Ice-ocean interactions are complicated, and a lot of interconnected processes occur in concert with each other. One repeated concern that I have in this paper is the tendency to over-state previous findings and their own results in the quest to find a clean and compelling story. By rewording the text in several places, the authors should acknowledge this complexity instead of simplifying it.

Authors : We would like to thank the reviewer for this detailed review, which contributed to improving the manuscript. We have accounted for each comment below as best as possible, in

order to describe our results as accurately as possible and avoid overstatements. In that regards, changes to the tone down the text, and acknowledge the complexity of ice-ocean interactions were made at L17, L38-43, L377, L433, L386-388, L434.

We have also added further description of the methodology at L480-516, L527-537, L548-549, L556-560, L583-586, L614-615, L629-632 and L658-661. You can find further information on these specific changes in the detailed response below.

detailed comments:

10: capitalize University

Authors: Agreed. Done.

13: This might seem totally pedantic, but ice shelves don't "drain" the ice sheet. The outlet glaciers do the "draining" – the ice shelves are the floating extensions of the ice shelves. Either reword that part of the sentence, or just delete that part of the sentence.

Authors: Agreed. We have reworded that part of the sentence accordingly, which now read in two parts: "We study the evolution of North Greenland glaciers that have long-considered to be a stable region. This part of Greenland hosts enough ice to raise sea level by 2.1 m, and is buttressed by the last remaining ice shelves of the ice sheet. »

14: capitalize Ice Sheet

Authors: Agreed. Done.

16: substantial and widespread increase in mass loss – Based on Figure 3, I'd call this moderate increase in mass loss. Steensby and 79N are the only really convincing ones.

Authors: Note that mass losses here are referring to ice shelf mass losses, rather than the ice discharge, which is what reviewer #2 is referring to at the moment with Figure 3. We however observe a widespread increase in basal melting (Fig 2) and ice shelf mass losses (Fig 3e). We have clarified this sentence which now read as "ice shelf mass losses" vs "mass losses" before. We have deleted the word "substantial" to avoid overstating the results.

25: Find original reference for this – IPCC synthesizes studies.

Authors: Agreed. We have supplemented our reference list with the study of Cáceres et al., 2020; Frederikse et al., 2020b

27-29: This is overstating the results of your 2018 paper, which focuses on glaciers in the SE, not "around Greenland"

Authors: We would like to bring to reviewer's 2 attention that this reference list includes a study led by Mike Wood, co-author of this study, and which focuses on all glaciers around Greenland. No change.

31-33: These last two sentences could use some reorganization – they seem contradictory and sloppily written.

Authors: Agreed. We have rephrased the sentence L30-33 in order to clarify the meaning, which is that North Greenland glaciers went out of balance in the early 2000s, which is ten years later than the greatest mass losers, their rates of mass losses through ice discharge is still moderate compared to other sectors.

36: which glaciers?

Authors: Agreed. We clarified the sentence at L36 by stating "the glaciers located in North Greenland".

36: None of those references actually calculate backstress at the grounding line. Instead, maybe compare the glacier basin size and potential SLR contribution.

Authors: Agreed. We have decided to clarify this paragraph by changing its structure. Now the paragraph starts by describing the potential SLR contribution, then the observed changes in ice shelves that have triggered increased mass losses through ice discharge. We have modified the references accordingly. We also would like to underline that we are already comparing glacier basin size at L70-78 (following paragraph), which is therefore answering Reviewer #2 suggestion.

39-41: I agree with this sentence – that it's important to understand the timing of calving and the glacier response. However, based on everything you've written before this sentence, it sounds like a solved problem. Revisit those references to highlight the complexity and uncertainty more accurately in this cause-effect of calving and acceleration.

Authors: Agreed. We rephrased the last two sentence of this paragraph L42-62 to better reflect this complexity. Now the sentence reads: "Despite their fundamental buttressing role, there is to date no comprehensive overview of these ice shelves evolution, which hampers our ability to understand the processes leading to their weakening and collapse, and their relation with glacier mass changes. It is thus extremely important to define the timing and drivers of historical and current changes of ice shelves, as well as glacier response, in order to better predict the contribution of Greenland to sea level rise."

41-42: I find this sentence confusing – I think you're referring to glaciers, not ice shelves.

Authors: This part has been clarified and rephrased in previous comment.

43: Buttressing has never really been quantified, and their role in glacier acceleration is a long-lasting debate in our field. Add references and/or acknowledge that this is a theoretical argument.

Authors: Disagreed. Buttressing index and buttressing response number has been quantified for ice shelves around Antarctica (see Fürst et al., 2016, and Reese et al., 2018), and also specifically for Petermann ice shelf (Rückamp et al., 2019). Moreover, the ABUMIP experiment, which gathered ice sheet experts internationally, has shown the crucial importance of ice shelf buttressing in the contribution of Antarctica to sea level rise (see Sun et al., 2020 for details). More specifically, it has demonstrated using multiple ice sheet models, that a complete collapse of all ice shelves around Antarctica would lead to a multi-meter increase in sea level rise within the first hundred years (Sun et al., 2020) . Finally, the role of ice shelf buttressing and impact on glacier stability has been observed with the collapse of the Larsen B, which triggers a drastic increase in the discharge of glaciers in the Antarctica peninsula (Rignot et al., 2004; Wuite et al., 2015), but also in the Amundsen sea embayment. To further illustrate my point, the evolution of ice shelf has been strongly highlighted by the IPCC reports (AR5+AR6+Special Report on the Ocean and Cryosphere) as one of the major scientific challenges if we want to better predict the future sea level rise.

46: In other places (e.g. 31) you capitalize "North"

Authors: Agreed. We have capitalized things throughout the text.

47: Are these SLE values what has already been contributed? Or their potential if glaciers accelerate?

Authors: These are Sea Level Equivalent values for each glacier basins. This is already specified at L71.

51: datasets

Authors: Done.

52: Conductivity Temperature Depth

Authors: Done.

53: ice shelves'

Authors: Done.

63: the GL "of" Zacharaie

Authors: Done.

65: delete "undocumented"

Authors: Done.

65: Abstract mentions four remaining ice shelves (16), as does Figure 2.

Authors: We have modified the abstract (L17), and the caption of Figure 2 for more consistency.

69: I'm confused by this sequence of sentences that include cumulative retreat and episodic frontal advance. Reword.

Authors: We have rephrased this sentence accordingly, which now reads L94: "In 2020, the ice front readvanced, leading to an increase of 35% of the floating area compared to 2012-2020."

70: When switching description to a new glacier, it helps the reader to put the glacier name at the start of the sentence, not the end. "At Ryder Glacier, between 1992 and 2011..."

Authors: Agreed. We have modified the text accordingly at L90-100.

65-81: In this paragraph, you refer to GL retreat in both area (73) and distance (74, 75, 77, 78). Keep it consistent and maybe add "average" distance to highlight across-width variations.

Authors: Agreed. We are now specifying in the text that we are using "average distance" to describe grounding line retreat (L93-108). In response to an earlier comment from a reviewer, we have made sure that we are describing grounding line changes both in term of retreat and ice shelf area.

84: remove ~ if you have uncertainties

Authors: Agreed. Done.

84: looks like Petermann melt values could have been higher before 2001, and lower again after 2016. Why compare an average of values over 3 yrs (2001-2004) with peak values in 2016? Same comparison is used for other glaciers too and seems a bit misleading or confusing.

Authors: Agreed. We have modified the paragraph accordingly in order to accurately describe the period when melt rates were higher L127-132.

93: twice as large

Authors: Done.

108: delete “hence compensating for thinning” (makes it sound like a physical compensation, not just a math compensation)

Authors: Done.

119: not statistically different

Authors: Without further details, it is impossible to guess what reviewer #2 want us to improve here. We wanted to note that this section has however been modified in response to a later comment of reviewer #2.

119: I don’t follow the “owing to calving fluxes” – how does that link to basal melt rates?

Authors: Ice shelves mass losses can be due to changes in mass from basal melting, Surface mass balance and calving flux. We have rephrased this sentence at L200.

123: I like how this is partitioned and clearly described – just add over what time period.

Authors: We have clarified this at L201 and 197.

141: Ice shelf changes were followed...

Authors: Done.

175: Consider rewording “tipping point” – as it is unclear what you mean here and it is a weighted term. What does it mean to reach its tipping point?

Authors: We agree with reviewer #2 that this part of the sentence is unclear. Consequently, we have preferred to remove it, in order to avoid confusions, as tipping point is a complex term, which can overstate the message of this sentence.

179: S1 shows an aerial image of Ostenfeld with mélange, then no mélange. Are there studies that quantify mélange buttressing support to an ice shelf (not just impacting calving rates to glacier termini)?

Authors: We are aware of studies that have shown buttressing effect of ice mélange on calving faces (Schlemm et al., 2021). We have added a reference to that paper at L348.

174-186: I’m a bit lost in this discussion paragraph – what’s the main point here? Reorganize and maybe add a concluding sentence to tie it together.

Authors: To avoid confusion we have removed the last sentence of the paragraph that was dealing with Strostrommen/Bistrup ice shelf (which has not collapsed). We have also added a concluding sentence on what would be needed to further investigate the processes leading to the collapse of these ice shelves L352-355. The point here was to provide a summary on the events that affected the ice shelves that collapse, the processes that have triggered these collapses and the role of ice shelves in buttressing the ice dynamics of upstream glaciers.

204: Add a bit more detail here – what model and what kind of runoff (surface?)

Authors: We are now specifying that this is model output from Regional atmospheric climate models and we are looking at surface runoff. This has been added at L382.

258: Quarterly? Does that mean 4x a year? I’m confused how quarterly is used in reference to GIMP DEMs between 2012 and 2016 (are those averaged?).

Authors: Yes, those are NSIDC DEM product, that were calculated by averaging DEM on quarter basis (3 months average). We have clarified the sentence accordingly at L439.

260: SAR DEMs over what time period?

Authors: We are now specifying the time period of the SAR DEMs (2016-2019) at L442.

270: Define what you're using for stable ground. What "ice" is considered stable? Was the distribution of the stable terrain relatively even?

Authors: We are now specifying what we mean as stable ground at L451. For the ice we use the same procedure as Shean et al., 2019. Since we use yearly averaged DEM, the observations from ASTER, GIMP and GLISTIN-A, should match the altimetry data we have from the same year. It is worth noting that the altimetry located on the ice has a relatively low impact on the coregistration since the vast majority of the reference DEM points are located on stable ground. Moreover, the location of stable terrain is even across all study domain. See examples below for Petermann, Ryder and 79N glacier on new Figure S39.

271: RMSE over stable terrain?

Authors: We have clarified the procedure accordingly at L451-471.

271: specify what version

Authors: We have specified the version accordingly at L468.

275: What do you mean by "coherency"?

Authors: We meant correlation score. We have more precisely described this procedure now at L452.

277: average standard deviation of what?

Authors: We now specify that this is the standard deviation of the difference with satellite altimetry at L455.

278: yearly difference of what?

Authors: We have also clarified this at L348.

279: you don't need to shift horizontally at all?

Authors: Yes we are also horizontally aligning DEMs twice. First when every single DEMs within a year are generated with MicMac, we coregister these single DEMs with a reference elevation map based on altimetry. After this, we stack all DEMs within a year to have the most comprehensive coverage of surface elevation data. This DEM is again aligned vertically and horizontally with the reference map. We have clarified our procedure at L451-471.

282: didn't have

Authors: Done.

282: why do you need to co-register the DEMs over ice? Isn't there enough stable terrain surrounding these glaciers to perform co-registration on land?

Authors: Since we use yearly averaged DEM, the observations from ASTER, GIMP and GLISTIN-A should match the altimetry data we have from the same year on the ice, if available. This is why we included values of ice surface elevation on the reference DEMs following the approach of Shean et al., 2019. The influence of altimetry on the ice has however a relatively low impact on the coregistration for North Greenland glaciers since the vast majority of the reference DEM points are located on stable ground. We have added Figure S40 to provide example of reference DEMs.

290: Reference Brunt et al (2010) and add some detail for uncertainties associated with assuming hydrostatic equilibrium near the grounding line. Brunt finds that equilibrium is found ~10 km after the grounding line for most ice shelves.

Authors: The paper of Brunt et al, compare different methods for deriving grounding zone location : using ICESat, a static DEM from the Mosaic of Antarctica, and InSAR. They found that using hydrostatic equilibrium hypothesis on a static DEM can yield to significant biases on the location of the landward limit of ice flexure from tidal movement. However this is not the method that we are using. We are deriving the location of the grounding zone using a dense mapping from Quadruple differential SAR Interferometry, which is in good agreement with ICESat derived grounding lines, and has the advantage of having a much better spatial coverage (as underlined by Brunt et al). We thanks the reviewer for this comment and have clarified this at L487-488.

307: GIMP v1 uses elevations from 1978 to 2009. Are you just using this topo correction over stable terrain or the full extent of the region? If for the whole area, won't there be biases from using a range of temporal datasets?

Authors: We use two different DEM to remove the topographic signal over the whole area. First GIMP v1, for ERC data (1992-2011), then GIMP v2 for Sentinel-1 (2014-2021). These two datasets were used because they provide a comprehensive, and high resolution coverage of the area. Moreover they are able to capture the surface elevation centered on the time period of both on the old data (ERS) and more recent observations (Sentinel-1) that are considered. However, it still assumes that no significant changes in ice surface elevation has occurred in the time interval between the DEM and SAR data. We have added a sentence to clarify this at L465-471..

308: Sentinel-1 (S1)

Authors: Done.

311: Doesn't this change with tides and will therefore have quite a spatial range?

Authors: Yes this is changing with tides. We have processed around 1000 interferograms for Petermann, 600 for Ryder, 520 for Steensby and 500 for 79N. More than 80% of this mapping comes from the Sentinel-1 dataset, which provides us with a good idea of the width of the grounding zone (change in grounding line position due to tides). We have supplemented and clarified the text accordingly at L508-515.

314: define "present"

Authors: Done.

316: reference for GEEDIT: <https://esurf.copernicus.org/articles/6/551/2018/esurf-6-551-2018.html>

Authors: Thanks. We have added the reference at L519.

334: Why isn't the same dataset used for shear strain as described in the paragraph above?

Authors: The calculation of strain rates is highly sensitive to the signal-to-noise ratio of flow velocity data. The annual mosaics are assembled from the same observations as described earlier, using a strict procedure of spatial and temporal filtering, thus allowing for the most accurate mosaic possible for strain rate calculations (Mouginot et al., 2017). We have clarified this in the text at line 537-541.

358: annual melt rates are reported in your figures above. Why isn't basal melt calculated

annually from the annual DEMs?

Authors: Calculating melt rates with a time difference of one year between DEMs requires having excellent vertical precision and horizontal resolution of the Digital Elevation Models (DEMs) used (Shean et al., 2019). Indeed, the shorter the time interval between the DEMs, the poorer the signal-to-noise ratio will be, and therefore inadequate for observing trends in melt rates. We conducted several tests with annual melt rate calculations, and the signal-to-noise ratio was insufficient to use these melt rates in our interpretations. Instead we used time difference >1 year between DEMs, and define the melt rate as the center date of each pairs (see L579). We have clarified this at L569-570.

363: annual

Authors: Done.

366: delete rapid

Authors: Done.

366: a third ice velocity product? I'm confused as to why different velocity products are used for each task.

Authors: No this is the same product as described before, we are sorry for the confusion, and clarified this at L450, were we now refer to the earlier paragraph on surface flow velocity.

375: Need references and some explanation for these errors of 15% of SMB and 100% for divergence.

Authors: We have clarified and added references for these errors at L460. Uncertainties on the SMB was determined conservatively using the comparison of Regional Atmospheric Climate Models with in-situ observations (Figure 5a of Fettweis et al., 2020). We specifically considered the highest range of differences (<-3mWE SMB values) which provides a conservative error on the SMB estimations in North Greenland.

379: Citation #5 only applies to Greenlandic glaciers below 66°N where CReSIS radar data is notoriously terrible and subsequently, so are BedMachine bed elevations. The depth radar collected above 76°N is actually much better.

Authors: We have tone down this sentence at L590. Note however that BedMachine v3 close to glacier grounding line is not only constrained by radar data but also by bathymetry measurements at the calving front, or at the glacier grounding line. As described in the Morlighem et al., 2017 study (which I participated into), the mass conservation algorithm is designed to converge to the available bathymetry data. Consequently, when the bathymetry is largely unknown, the uncertainty of Bedmachine at the grounding line is high (Morlighem et al., 2017). This is typically the case for Petermann, Ryder, Steensby and 79N where no bathymetry data are available (see Fig S18).

388: The calving flux methodology seems like it might need some revisiting. How “off” are the calving fluxes? How frequently and of what magnitude are the positive fluxes? Pinning all the discrepancy over SMB (presumably that's SMB over the shelf), doesn't seem like it would account for it, as it is likely small. This requires a bit more digging in to.

Authors: Agreed. We have revised our approach and removed that part on putting the discrepancy over SMB, which resulted in abnormally high SMB when ice front readvanced. Instead we have decided to keep positive calving values, but to label them, as mass gain throughout frontal advance or grounding line retreat. This allowed us to add Bistrup and

Storstrommen into our ice shelf mass balance partitioning analysis. This was described at L493-494. We adjusted Figure 3, and the description of the partitioning accordingly.

426: what box?

Authors: We have added a reference to Figure S17 where the boxes locations appear.

Figures:

Figure 1:

- Move color bars for grounding line and ice front to top of figure (or at least above the map), so closer to the relevant panels.
- Not sure you need logarithmic ice velocity, since you aren't highlighting slow flow.
- I can't see the flux gates for E-H
- Ice shelf colors correspond....

Authors: Agreed, we have removed the logarithmic scale, and used a linear one. We only show flux gates for ice shelves that where we discuss the discharge (Fig 3). We have moved the legend in the center of the figure having, which makes it more centralized towards the maps these are related to. We have also corrected the typo underlined by reviewer #2 inside the caption. We have also changed our way to display the partitioning in this figure using a bar plot, which also allows us to include the SMB component, since we slightly adjusted our partitioning analysis in response to an earlier comment.

Figure 2:

- I'm confused by the black dashed line – does it pertain to a specific year, or is it just the length over which basal melt rates are calculated? Is that what goes into the panels on the right? Why different distances?

Authors: Yes this is the length over which the average basal melt rate on the right is calculated. We adapted the length of this distance on a case by case basis to focus specifically on the grounding line region where the highest melt rates are displayed. We clarified the sentence accordingly inside the caption.

- Add “Note change in basal melt scale bar for each panel”

Authors: Done.

- Add lines, or bold years where major calving events occurred

Authors: This is already noted in Figure 3, where melt rates also appear. It would be doubling the info so we choose to keep calving events in Figure 3.

Figure 3:

- Glacier dynamical response to ice shelf changes (or, if keeping original phrasing, add apostrophes)

Authors: Done.

References are inconsistent and several are incomplete (#8, #30)

Authors: Done.

Reviewers' Comments:

Reviewer #2:

Remarks to the Author:

Authors edits substantially improved the manuscript - thank you for considering them.

Response to reviewers

REVIEWERS' COMMENTS

Reviewer #2 (Remarks to the Author):

Authors edits substantially improved the manuscript - thank you for considering them.

Authors: We would like to thank both Reviewer #1 and #2 for providing insightful comments, which contributed to greatly improve the quality of our manuscript.